# Denoising and Alignment: Rethinking Domain Generalization for Multimodal Face Anti-Spoofing

## Abstract

Face anti-spoofing is essential for securing face recognition in applications such as payments, border control, and surveillance. Current multimodal methods often degrade under domain shift and modality bias. We address these challenges with a Multimodal Denoising and Alignment framework (MMDA) built around two threads, denoising and alignment. Using a pretrained CLIP backbone, the Modality–Domain Joint Differential Attention (MD2A) module suppresses modality noise and domain noise at fusion to produce cleaner representations that lay the groundwork for alignment. The Representation Space Soft alignment (RS2) then maps the fused representation to text-defined class subspaces rather than a single prompt, preserving semantics while improving class separability and cross-domain consistency. Finally, the U-shaped Dual Space Adaptation (U-DSA) applies alignment across layers and feeds deep information back to shallow layers, preserving pretrained semantics while adding task-specific capacity. The three components act jointly: denoising stabilizes alignment, alignment tightens decision boundaries, and U-DSA consolidates and propagates these gains, yielding stronger multimodal domain generalization. MMDA attains state-of-the-art results on four public datasets under multiple protocols: under the complete modality setting, it reduces HTER by 9.63% and increases AUC by 5.98% over the strongest prior.

## 1 Introduction

Facial recognition (FR) systems are critical in authentication contexts such as payment processing, identity verification, surveillance, and attendance tracking, emphasizing the need for robust security measures Yu et al. (2022); Xu et al. (2023). However, FR systems are vulnerable to presentation attacks, which can lead to false identifications through tactics like printed photographs, video playbacks, and 3D masks, posing significant risks to the financial, transportation, and safety sectors. Consequently, numerous Face Anti-Spoofing (FAS) methods have been proposed Cai et al. (2024b); Yu et al. (2024c); Jiang et al. (2024) to address these security challenges.

With advancements in multimodal learning and sensor manufacturing, multi-modal FAS has been widely applied in real-world scenarios, commonly using RGB, Depth, and infrared sensors. Compared to single-modal FAS, multi-modal FAS can obtain more useful information, such as spatial geometric and temperature information, allowing for more comprehensive and accurate modeling and the extraction of richer deception cues. One significant challenge for FAS is poor generalizability, particularly performance degradation when encountering domain shifts and unseen attacks. Many domain generalization (DG) methods have been proposed to address this issue using techniques like domain alignment, feature disentanglement, and adversarial training. However, these DG methods are designed for unimodal FAS and do not yield satisfactory results when directly transferred to multimodal scenarios Lin et al. (2024). Existing FAS methods overlook the fact that multimodal performance and domain generalization performance, although interrelated in final outcomes, have distinct underlying principles: multimodal performance relies on good alignment and sufficient modality interaction, while domain generalization performance depends on learning domain-invariant information Dong et al. (2023).

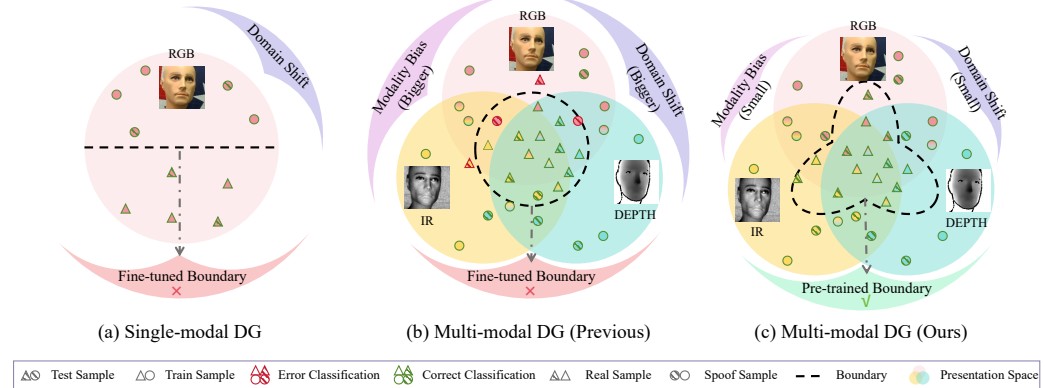

(a) Single-modal DG      (b) Multi-modal DG (Previous)      (c) Multi-modal DG (Ours)

Figure 1: (a) In the single-modal FAS scenario, the presence of domain shifts leads to domain generalization issues. (b) In the multi-modal FAS scenario, the existence of modality biases causes the gap between the infrared and depth modalities to be significantly larger than that between RGB modalities. The combined effect of modality biases and domain shifts amplifies noise, making multi-modal FAS more challenging. (c) Our proposed method not only reduces noise but also avoids overly smooth decision boundaries, thereby alleviating the issue of test samples with severe domain shifts failing to be correctly distinguished. **Note:** Face images are from the WMCA George et al. (2019) dataset.

In FAS, as shown in Fig. 1 (b), multi-modal DG may be more challenging than single-modal DG due to: **(1) More diverse noises:** Unimodal scenarios face domain shifts due to sensor, lighting, and other factors, introducing domain noise and increasing feature distribution divergence across domains. Multimodal data also encounter modality bias, with differences in sensors and imaging principles introducing modality noise Dong et al. (2023). When both noises coexist, feature differences in multimodal data across domains become more pronounced, exacerbating domain shifts. **(2) More complex alignments:** The unpredictable nature of domain shifts in multimodal combinations makes alignment more complex in DG. Decision boundaries learned through carefully designed modules may fail to adapt to the complex representations between modalities in different domains.

To address the first issue, we propose an improved attention fusion module for a unified denoising strategy. Inspired by feature denoising Ye et al. (2024); Dong et al. (2023), we extract common noise features from multimodal samples within the same domain to improve the attention mechanism. By performing a differential operation between noisy and pure noise features, we suppress the attention module's focus on noise, enabling the model to concentrate on effective information. As shown in Fig. 1 (c), this strategy can handle both domain noise and modality noise simultaneously, avoiding the complexity brought by specific module processing and enhancing the capability of multimodal data processing.

Regarding issue (2), instead of directly learning a generalized decision boundary, we construct a generalized representation space and map data into this space, maintaining the pre-trained model's representation space boundaries to reduce overfitting risk. CLIP's cross-modal contrastive learning capabilities make it suitable for building a generalized representation space. We use pre-trained CLIP to align multi-domain multimodal data into this space with the help of text. However, CLIP's focus on visual-text alignment can weaken visual modality representations. Therefore, we propose a relaxed soft alignment scheme, allowing for flexible alignment and preventing representation weakening. We also design a module to protect and adjust the representation during alignment, optimizing the results and enhancing the model's multimodal performance and generalization capability. To sum up, our contributions include:

- We propose a CLIP-based multimodal FAS framework, namely **M**ulti**m**odal **D**enoising and **A**lignment (**MMDA**), which possesses exceptional cross-domain generalization capabilities.

- Within MMDA, we propose **M**odality-**D**omain Joint **D**ifferential **A**ttention (**MD2A**), which identifies and eliminates modality noise and domain noise from images to learn generalized multimodal representations.

- To further enhance generalization, we design the **R**epresentation **S**pace **S**oft (**RS2**) Alignment, which, with its flexible alignment constraints, effectively preserves complex representations in the generalized representation space. Moreover, we design a **U**-shaped **D**ual **S**pace **A**daptation (**U-DSA**) module to enhance the adaptability of representations while maintaining generalization performance. These two improvements not only enhance the framework's generalization capabilities but also boost its ability to represent complex representations.

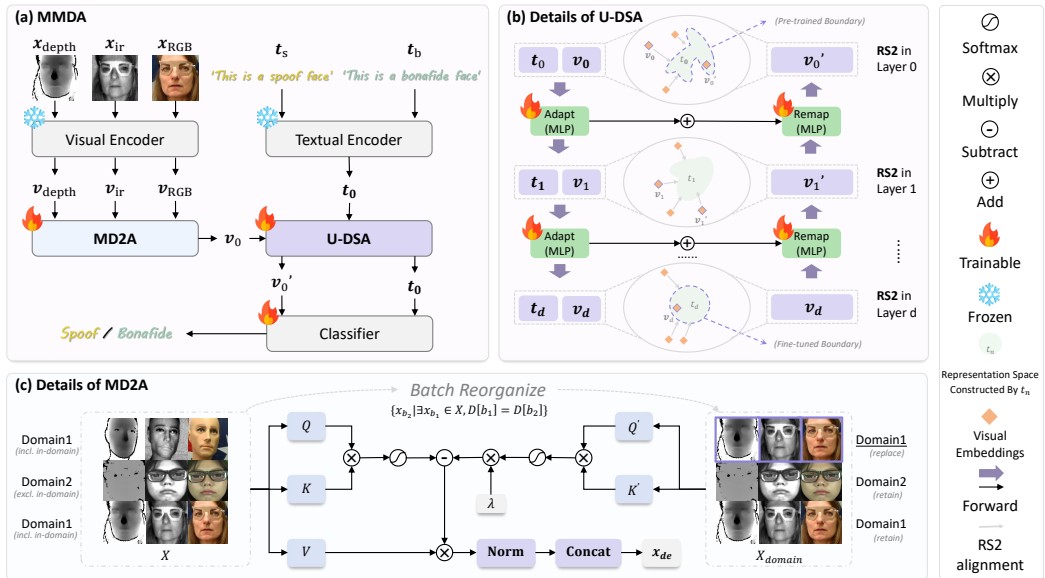

Figure 2: Overall framework of the proposed MMDA. (a) Overall process of MMDA. (b) Details of the U-shaped Dual Space Adaptation (U-DSA) module and the application method of the Representation Space Soft (RS2) alignment approach. (c) Operational details of the Modality-Domain Joint Differential Attention (MD2A). **Note:** All face images shown are from the WMCA George et al. (2019), CeFA Liu et al. (2021a), PADISI Rostami et al. (2021), and SURF Zhang et al. (2020) datasets.

- Our extensive experimental evaluations affirm that the MMDA Framework has achieved state-of-the-art (SOTA) results across a spectrum of evaluation protocols and benchmark tests.

## 2 RELATED WORKS

### 2.1 FACE ANTI-SPOOFING

In the field of FAS, deep learning has led to the development of numerous architectures for extracting discriminative spoofing cues to distinguish live from fake faces. Despite impressive performance in known domains, FAS performance severely degrades under domain shifts in unknown domains (e.g., changes in lighting and sensor types) Yu et al. (2022); Huang et al. (2022). To enhance practicality, recent efforts have focused on improving domain generalization capabilities, using techniques like adversarial learning Jiang et al. (2023); Yue et al. (2023), feature disentanglement Liu et al. (2024b); Jiang et al. (2024); Zhou et al. (2023), meta-learning Cai et al. (2022); Du et al. (2022), data augmentation Cai et al. (2024a); Ge et al. (2025), and domain alignment Hu et al. (2024); Wang et al. (2024); Le & Woo (2024). These aim to extract domain-invariant features for more generalizable decision boundaries. However, most methods are designed for unimodal scenarios and struggle to integrate multimodal information effectively, leading to suboptimal generalization.

Multimodal FAS integrates data from RGB, depth, and infrared to detect live and spoofed faces, leveraging unique information from each modality Liu et al. (2023); Kong et al. (2022; 2024); Yu et al. (2020); Lin et al. (2024). Recent studies have used attention-based fusion and adaptive loss functions to extract complementary information George & Marcel (2021); Zheng et al. (2025), and cross-modal translation to address semantic differences Liu et al. (2021b). Recently, numerous studies have explored multimodal FAS under conditions with missing modality inputs and proposed protocols and methods to enhance robustness Yu et al. (2024a;b; 2023). To enable flexible FAS under various modality combinations, cross-modal attention and multimodal adapters with pre-trained ViT are used to learn modality-insensitive features, improving generalization Liu & Liang (2023); Liu et al. (2023); Lin et al. (2024). However, these methods mainly focus on multimodal performance, often overlooking the complex domain generalization challenges in multimodal settings.

### 2.2 PARAMETER-EFFICIENT TRANSFER LEARNING

Parameter-efficient transfer learning (PETL) adapts large pre-trained models like Vision Transformers (ViT) Dosovitskiy (2020) and CLIP Radford et al. (2021) to new domains by fine-tuning a small subset of parameters, reducing overfitting and training costs while maintaining generalization. For FAS task, PETL has shown significant performance Cai et al. (2023; 2024b); Srivatsan et al. (2023). For example, S-Adapter Cai et al. (2024b) uses lightweight modules to adjust pre-trained features, and SA-FAS Sun et al. (2023) enhances PETL through improved training strategies and loss functions. Using CLIP as the backbone Liu et al. (2024a); Srivatsan et al. (2023); Fang et al. (2024); Liu et al. (2025), text prompts enhance generalization Gao et al. (2024), providing context and semantic guidance to improve model robustness in complex FAS scenarios. However, existing methods focus mainly on the generalization of pre-trained model weights, paying less attention to the generalization representations of the pre-trained space, which can enhance models' transfer and adaptation in new tasks.

---

**Algorithm 1:** Modality-Domain Joint Differential Attention

**Input:** batch samples $X = \{\boldsymbol{x}_0, \boldsymbol{x}_1, \ldots, \boldsymbol{x}_b\}$; domain labels $\mathcal{D} = \{\boldsymbol{d}_0, \boldsymbol{d}_1, \cdots, \boldsymbol{d}_b\}$

**Output:** denoised samples $X_{denoise}$

1 # Batch Reorganize
2 **for** $i \leftarrow 0$ *to* $b$ **do**
3     **for** $j \leftarrow 0$ *to* $b$ **do**
4        # Locate samples from the same domain.
5        **if** $\mathcal{D}[i] = \mathcal{D}[j]$ **then**
6           # Concat facilitates subsequent computations.
7           $X[i] \leftarrow [\, X[i];\, X[j]\,]$;
8           **break**;
9        **end**
10     **end**
11 **end**
12 # Split features and noise
13 $(Q,\, Q_{\text{noise}}) \leftarrow \texttt{split}(XW_q)$;
14 $(K,\, K_{\text{noise}}) \leftarrow \texttt{split}(XW_k)$;
15 $V \leftarrow XW_v$;
16 # $n_d$ is the feature dimension
17 $s \leftarrow 1/\sqrt{n_d}$;
18 $A \leftarrow (Q\, K^\top) \cdot s$;
19 $A_{\text{noise}} \leftarrow (Q_{\text{noise}}\, K_{\text{noise}}^\top) \cdot s$;
20 # Denoising
21 $X_{denoise} \leftarrow \big(\texttt{softmax}(A) - \lambda\, \texttt{softmax}(A_{\text{noise}})\big)\, V$;

---

## 3 METHODOLOGY

Our MMDA framework is illustrated in Fig. 2 (a). Initially, input images and captions are processed through the CLIP backbone network to obtain embedding vectors. The visual embedding vectors are denoised and modality-fused via the proposed Modality-Domain Joint Differential Attention (MD2A), then aligned using the Representation Space Soft Alignment (RS2) method, and adjusted with the U-shaped Dual Space Adaptation (U-DSA) module. Finally, these processed visual embedding vectors are combined with the text embedding vectors to participate in the classification process of the classifier. For a comparison of the classification methods and details during training and testing, please refer to Appendix B.

### 3.1 PRELIMINARY

The CLIP pre-trained model is known for its outstanding zero-shot performance, with a richly generalized embedding space. CLIP Radford et al. (2021) includes an image encoder and a text encoder. In FAS, it uses textual prompts for real and fake face descriptions. CLIP classifies images by computing similarity to these prompts, selecting the highest-scoring category. After standard fine-tuning Srivatsan et al. (2023), CLIP performs well in face anti-spoofing. However, CLIP's focus on visual-text alignment lacks constraints for visual-visual alignment in multimodal domain generalization, potentially neglecting the generalization of visual representations. Specifically, CLIP may not adequately capture and align subtle differences in visual features across modalities, affecting its cross-domain generalization capability.

### 3.2 DENOISING OF MODALITY NOISE AND DOMAIN NOISE

As shown in Fig. 2 (c), to effectively eliminate domain noise and modality noise for obtaining a more reliable representation, our proposed Modality-Domain Joint Differential Attention (MD2A) mechanism first randomly selects different or the same data within the same domain from the input multi-domain dataset $X = \{\boldsymbol{x}_0, \boldsymbol{x}_1, \ldots, \boldsymbol{x}_b\}$ to construct a domain sample set $X_{\text{domain}}$. This step helps capture the noise characteristics within the domain, as the differences between different data points can reveal the patterns of noise. The overall process is shown in Alg. 1, specifically, for each sample $\boldsymbol{x}_{b_1}$, we find another sample $\boldsymbol{x}_{b_2}$ within the same domain, denoted as $\boldsymbol{x}'_{b_1} = \boldsymbol{x}_{b_2}$, where

$Domain[b_1] = Domain[b_2]$ ensures that the samples are from the same domain. This is represented as follows:

$$X_{\text{domain}} = \{\boldsymbol{x}_{b_2} \mid \exists \boldsymbol{x}_{b_1} \in X, \ \mathcal{D}[\boldsymbol{x}_{b_1}] = \mathcal{D}[\boldsymbol{x}_{b_2}]\}, \tag{1}$$

where $\mathcal{D}$ denotes domain. Next, the algorithm extracts domain noise from $X_{\text{domain}}$ and calculates the domain noise attention weights $A'$. This is achieved by multiplying the input data $X$ with query weights $W_q$ and key weights $W_k$ and then splitting them into two parts, namely $(Q, Q') = \text{split}(XW_q)$ and $(K, K') = \text{split}(XW_k)$. Meanwhile, the value $V$ is calculated as $V = XW_v$, where $W_v$ is the value weight matrix.

The calculation of attention weights accounts for feature dimension scaling, represented as $s = 1/\sqrt{n_d}$, where $n_d$ is the dimension of the features. The denoised sample $X_{\text{denoise}}$ is computed through the following integrated formula:

$$X_{\text{denoise}} = \left[\text{softmax}\left(\frac{QK^\top}{\sqrt{n_d}}\right) - \lambda \cdot \text{softmax}\left(\frac{Q'K'^\top}{\sqrt{n_d}}\right)\right] V, \tag{2}$$

where $\lambda$ is a tuning parameter that balances the influence of the two attention mechanisms.

This method adaptively mitigates domain noise effects, yielding stable denoised data through dynamic weight adjustment, ensuring model robustness against varied noise patterns. Domain differential attention, an extension of differential attention, handles both domain and modality noise simultaneously in multi-modal data. When $X_{\text{domain}}$ matches $X$, it functions as differential attention. However, in multi-modal contexts where $X$ and $X_{\text{domain}}$ encompass both domain and modality information, this approach addresses sensor and environmental noise, enhancing model generalization and robustness in multi-modal tasks.

### 3.3 REPRESENTATION SPACE ALIGNMENT

**Representation Space Soft Alignment.** As shown in Fig. 2(b), RS2 aligns the fused visual embedding to *text defined class subspaces* instantiated by caption sets rather than a single prompt. We prepare two caption collections $C_{\text{live}}$ and $C_{\text{spoof}}$ (Appendix A) and obtain the corresponding text embedding sets via the CLIP text encoder:

$$T_{\text{live}} = \left\{\boldsymbol{t}_{\text{live}}^{(k)}\right\}_{k=1}^{K}, \quad T_{\text{spoof}} = \left\{\boldsymbol{t}_{\text{spoof}}^{(k)}\right\}_{k=1}^{K}, \quad T = T_{\text{live}} \cup T_{\text{spoof}}. \tag{3}$$

Let $V = \{\boldsymbol{v}_i\}$ denote fused visual embeddings. RS2 measures the proximity between a visual embedding and the text-defined subspace using the nearest-anchor cosine distance *scaled to* $[0, 1]$:

$$\boldsymbol{d}_i = \frac{1}{2} \min_{\boldsymbol{t} \in T}\left(1 - \frac{\boldsymbol{v}_i \cdot \boldsymbol{t}}{\|\boldsymbol{v}_i\| \|\boldsymbol{t}\|}\right) \in [0, 1]. \tag{4}$$

The alignment loss then encourages visual embeddings to approach the region associated with their class:

$$\mathcal{L}_{\text{align}} = -\sum_{\boldsymbol{v}_i \in V}\left[\boldsymbol{y}_i \log(1 - \boldsymbol{d}_i) + (1 - \boldsymbol{y}_i) \log(\boldsymbol{d}_i)\right], \tag{5}$$

where we set $\boldsymbol{y}_i = 0$ for *spoof* and $\boldsymbol{y}_i = 1$ for *live*.

To stabilize the decision boundary, a shared linear head is trained with both visual and text embeddings. Denote by $p_{\boldsymbol{e}}$ the spoof probability predicted for an embedding $\boldsymbol{e}$. With $\boldsymbol{y}_j$ the class label of $\boldsymbol{e}_j$, the classification loss is

$$\mathcal{L}_{\text{cls}} = -\sum_{\boldsymbol{e}_j \in \{V, T\}}\left[\boldsymbol{y}_j \log(1 - p_{\boldsymbol{e}_j}) + (1 - \boldsymbol{y}_j) \log(p_{\boldsymbol{e}_j})\right]. \tag{6}$$

The final RS2 objective combines the two terms:

$$\mathcal{L}_{\text{RS2}} = \mathcal{L}_{\text{cls}} + \mathcal{L}_{\text{align}}. \tag{7}$$

During training, captions instantiate the class subspaces and supervise alignment; for the classifier based route reported in the main tables, test time does not require text input (Appendix B).

Table 1: Cross-dataset testing results under the fixed-modal scenarios (Protocol 1) among CASIA-CeFA (C), PADISI (P), CASIA-SURF (S), and WMCA (W). Best results are marked in **bold**.

| Method | CPS→W | | CPW→S | | CSW→P | | PSW→C | | Average | |
|---|---|---|---|---|---|---|---|---|---|---|
| | HTER(%)↓ | AUC(%)↑ | HTER(%)↓ | AUC(%)↑ | HTER(%)↓ | AUC(%)↑ | HTER(%)↓ | AUC(%)↑ | HTER(%)↓ | AUC(%)↑ |
| Uni-modal DG (Concat + 1*1 Conv) | | | | | | | | | | |
| SSDG Jia et al. (2020) | 26.09 | 82.03 | 28.50 | 75.91 | 41.82 | 60.56 | 40.48 | 62.31 | 37.32 | 68.25 |
| SSAN Wang et al. (2022) | 17.73 | 91.69 | 27.94 | 79.04 | 34.49 | 68.85 | 36.43 | 69.29 | 35.34 | 70.98 |
| SA-FAS Sun et al. (2023) | 21.37 | 87.65 | 23.22 | 84.49 | 35.10 | 70.86 | 35.38 | 69.71 | 28.77 | 78.18 |
| IADG Zhou et al. (2023) | 27.02 | 86.50 | 23.04 | 83.11 | 32.06 | 73.83 | 39.24 | 63.68 | 39.83 | 62.95 |
| FLIP Liu & Liang (2023) | 13.19 | 93.79 | 11.73 | 94.93 | 17.39 | 90.63 | 22.14 | 83.95 | 16.11 | 90.83 |
| Multi-modal FAS | | | | | | | | | | |
| ViT Dosovitskiy (2020) | 20.88 | 84.77 | 44.05 | 57.94 | 33.58 | 71.80 | 42.15 | 56.45 | 36.60 | 68.12 |
| AMA Yu et al. (2024b) | 17.56 | 88.74 | 27.50 | 80.00 | 21.18 | 85.51 | 47.48 | 55.56 | 27.47 | 79.85 |
| VP-FAS Yu et al. (2024a) | 16.26 | 91.22 | 24.42 | 81.07 | 21.76 | 85.46 | 39.35 | 66.55 | 29.82 | 76.62 |
| ViTAF Huang et al. (2022) | 20.58 | 85.82 | 29.16 | 77.80 | 30.75 | 73.03 | 39.75 | 63.44 | 33.89 | 71.54 |
| MM-CDCN Yu et al. (2020) | 38.92 | 65.39 | 42.93 | 59.79 | 41.38 | 61.51 | 48.14 | 53.71 | 46.81 | 53.43 |
| CMFL George & Marcel (2021) | 18.22 | 88.82 | 31.20 | 75.66 | 26.68 | 80.85 | 36.93 | 66.82 | 31.01 | 75.07 |
| MMDG Lin et al. (2024) | 12.79 | 93.83 | 15.32 | 92.86 | 18.95 | 88.64 | 29.93 | 76.52 | 22.93 | 84.19 |
| DADM Yang et al. (2025) | 11.71 | 94.89 | 6.92 | 97.66 | 19.03 | 88.22 | 16.87 | 91.08 | 13.63 | 92.96 |
| CLIP Radford et al. (2021) | 14.55 | 90.47 | 18.17 | 90.02 | 24.13 | 83.15 | 38.33 | 65.71 | 24.63 | 83.00 |
| **MMDA (Ours)** | **1.22** | **99.99** | **4.21** | **98.62** | **4.34** | **98.58** | **6.25** | **98.18** | **4.00** | **98.94** |

Table 2: Cross-dataset testing results under the missing modalities scenarios (Protocol 2) among CASIA-CeFA (C), PADISI (P), CASIA-SURF (S), and WMCA (W). Best results are marked in **bold**.

| Method | Missing D | | Missing I | | Missing D & I | | Average | |
|---|---|---|---|---|---|---|---|---|
| | HTER(%)↓ | AUC(%)↑ | HTER(%)↓ | AUC(%)↑ | HTER(%)↓ | AUC(%)↑ | HTER(%)↓ | AUC(%)↑ |
| Uni-modal DG (Concat + 1*1 Conv) | | | | | | | | |
| SSDG Jia et al. (2020) | 38.92 | 65.45 | 37.64 | 66.57 | 39.18 | 65.22 | 38.58 | 65.75 |
| SSAN Wang et al. (2022) | 36.77 | 69.21 | 41.20 | 61.92 | 33.52 | 73.38 | 37.16 | 68.17 |
| SA-FAS Sun et al. (2023) | 36.30 | 69.07 | 39.80 | 62.69 | 33.08 | 74.29 | 36.40 | 68.68 |
| IADG Zhou et al. (2023) | 40.72 | 58.72 | 42.17 | 61.83 | 37.50 | 66.90 | 40.13 | 62.49 |
| FLIP Liu & Liang (2023) | 23.66 | 83.90 | 24.06 | 84.04 | 27.07 | 79.79 | 27.93 | 79.44 |
| Multi-modal FAS | | | | | | | | |
| ViT Dosovitskiy (2020) | 40.04 | 64.69 | 36.77 | 68.19 | 36.20 | 69.02 | 37.67 | 67.30 |
| AMA Yu et al. (2024b) | 29.25 | 77.70 | 32.30 | 74.06 | 31.48 | 75.82 | 31.01 | 75.86 |
| VP-FAS Yu et al. (2024a) | 29.13 | 78.27 | 29.63 | 77.51 | 30.47 | 76.31 | 29.74 | 77.36 |
| ViTAF Huang et al. (2022) | 34.99 | 73.22 | 35.88 | 69.40 | 35.89 | 69.61 | 35.59 | 70.64 |
| MM-CDCN Yu et al. (2020) | 44.90 | 55.35 | 43.60 | 58.38 | 44.54 | 55.08 | 44.35 | 56.27 |
| CMFL George & Marcel (2021) | 31.37 | 74.62 | 30.55 | 75.42 | 31.89 | 74.29 | 31.27 | 74.78 |
| MMDG Lin et al. (2024) | 24.89 | 82.39 | 23.39 | 83.82 | 25.26 | 81.86 | 24.51 | 82.69 |
| DADM Yang et al. (2025) | 21.56 | 85.17 | 20.82 | 85.28 | 22.61 | 84.04 | 21.66 | 84.83 |
| CLIP Radford et al. (2021) | 28.07 | 77.00 | 29.10 | 77.04 | 32.58 | 73.36 | 33.83 | 71.11 |
| **MMDA (Ours)** | **11.10** | **93.97** | **5.98** | **98.30** | **13.36** | **93.74** | **10.14** | **95.33** |

**U-shaped Dual Space Adaptation Module.** As shown in Fig. 2 (b), when fine-tuning pre-trained weights for multimodal FAS, increasing the depth of the fine-tuning module is common to enhance feature extraction Zhou et al. (2024). However, downstream task data often lacks generalization information compared to pre-training data, causing the module to focus excessively on task-specific features and weaken generalizable feature extraction. This results in a decline in generalization performance, with decision boundaries smoothing in the representation space. While aligning data to the pre-trained representation space maintains some generalization, the lack of deep structure limits feature extraction. Conversely, increasing module depth improves feature extraction but can smooth decision boundaries, weakening generalization and increasing overfitting risk. Clearly, relying solely on layer design and deep module improvements cannot fundamentally solve this problem.

To address this, we propose the U-shaped Dual Space Adaptation (U-DSA) module with two key points: First, RS2 alignment at each layer ensures semantic relationship optimization between modalities. Secondly, we remap the visual embedding $v'_{i-1}$ of the deeper layer back to the shallower space of the previous layer and perform a residual operation with $v_i$ to obtain $v'_i$, thereby enhancing generalizable representations. Specifically, assuming the maximum number of layers in U-DSA is $d$ and the current layer is $i$, this process can be formulated as:

$$v'_i = \underbrace{\text{Adapt}(v_{i-1})}_{\text{Equal to } v_i} + \text{Remap}(v'_{i+1}), \quad (8)$$

where $i, d \in \mathbb{N}$ and $v'_i$ represents the enhanced embedding after the residual operation. When $i = 0$, $v_0$ is directly provided by the input, not via Adapt from $v_{i-1}$. When $i = d$, $v'_d$ is the deepest layer's output, skipping Remap. The operations of Adapt and Remap are implemented by simple MLP. This design leverages residual connections to feedback deep features to the shallow space, enhancing generalization while fully utilizing deep-layer processing. The U-shaped structure avoids intermediate layers between the representation space and classifier, reducing generalization loss and fully processing visual embeddings. This allows us to fully utilize the decision boundaries of the pre-trained representation space, thereby enhancing generalization capability.

Table 3: Cross-dataset testing results under the limited source domain scenarios (Protocol 3) among CeFA (C), PADISI (P), SURF (S), and WMCA (W). The best results are in **bold**.

| Method | CW→PS | | PS→CW | |
|---|---|---|---|---|
| | HTER(%)↓ | AUC(%)↑ | HTER(%)↓ | AUC(%)↑ |
| Uni-modal DG (Concat + 1*1 Conv) | | | | |
| SSDG Jia et al. (2020) | 25.34 | 80.17 | 46.98 | 54.29 |
| SSAN Wang et al. (2022) | 26.55 | 80.06 | 39.10 | 67.19 |
| SA-FAS Sun et al. (2023) | 25.20 | 81.06 | 36.59 | 70.03 |
| IADG Zhou et al. (2023) | 22.82 | 83.85 | 39.70 | 63.46 |
| FLIP Liu & Liang (2023) | 15.92 | 92.38 | 23.85 | 83.46 |
| Multi-modal FAS | | | | |
| ViT Dosovitskiy (2020) | 42.66 | 57.80 | 42.75 | 60.41 |
| AMA Yu et al. (2024b) | 29.25 | 76.89 | 38.06 | 67.64 |
| VP-FAS Yu et al. (2024a) | 25.90 | 81.79 | 44.37 | 60.83 |
| ViTAF Huang et al. (2022) | 29.64 | 77.36 | 39.93 | 61.31 |
| MM-CDCN Yu et al. (2020) | 29.28 | 76.88 | 47.00 | 51.94 |
| CMFL George & Marcel (2021) | 31.86 | 72.75 | 39.43 | 63.17 |
| MMDG Lin et al. (2024) | 20.12 | 88.24 | 36.60 | 70.35 |
| DADM Yang et al. (2025) | 12.61 | 93.81 | 20.40 | 89.51 |
| CLIP Radford et al. (2021) | 19.36 | 90.57 | 29.98 | 79.22 |
| **MMDA (Ours)** | **7.52** | **96.84** | **6.30** | **98.35** |

Table 4: Ablation results on the proposed Modality-Domain Joint Differential Attention (MD2A).

| Method | HTER(%)↓ | AUC(%)↑ |
|---|---|---|
| Dense Adaptor | 23.26 | 84.92 |
| Dense Adaptor (w/ MHSA) | 25.85 | 82.95 |
| Dense Adaptor (w/ DA) | 16.49 | 92.05 |
| **Dense Adaptor (w/ MD2A)** | **13.47** | **94.20** |
| MoE Adaptor | 22.92 | 85.84 |
| MoE Adaptor (w/ MHSA) | 12.83 | 93.25 |
| MoE Adaptor (w/ DA) | 12.72 | 93.89 |
| **MoE Adaptor (w/ MD2A)** | **9.70** | **95.23** |

Table 5: Ablation results on RS2 Alignment.

| Method | HTER(%)↓ | AUC(%)↑ |
|---|---|---|
| Vanilla Alignment | 9.70 | 95.23 |
| Smooth Alignment | 9.17 | 96.32 |
| **RS2 Alignment** | **8.88** | **97.20** |

## 4 EXPERIMENT

We follow the MMDG protocol Lin et al. (2024) across complete modalities, missing modalities at test time, and limited source domains; detailed splits and rules are in Appendix E. Experiments use WMCA (**W**) George et al. (2019), CeFA (**C**) Liu et al. (2021a), PADISI (**P**) Rostami et al. (2021), and SURF (**S**) Zhang et al. (2020). Metrics are HTER and AUC. Unless noted, main tables report the classifier based inference route; the zero-shot variant and the training–testing comparison are in Appendix B.

**Implementation Details.** We use a CLIP visual encoder with $224 \times 224$ inputs and 16 pixel patches ($14 \times 14$ token grid) and project the global embedding to 512 dimensions; CLIP is pretrained on large scale image–text pairs. Images follow CLIP preprocessing; face cropping follows each dataset. Training uses AdamW with learning rate $5 \times 10^{-6}$, weight decay $1 \times 10^{-3}$, batch size 24, and 80 epochs. The alignment loss and classification loss are summed with equal weights. The U-DSA depth is 7. Caption sets are listed in Appendix A. Computational cost and timing are reported in Appendix D.

### 4.1 CROSS-DATASET TESTING

**Complete Modality Scenario.** Protocol 1 is designed to evaluate model performance across unseen domains using multimodal data from varied scenarios. For example, the sub-protocol **CPS →** **W** represents that we take **C**, **P**, and **S** as training sets, while **W** is testing set. As shown in Table 1, our method achieved the best results across all sub-protocols. Specifically, the average HTER was 4.00%, which is 9.63% lower than the second-best method; the AUC was 98.94%, which is 5.98% higher than the second-best method. Moreover, the metrics for all sub-protocols were very close to perfect accuracy. These results strongly corroborate our analysis of the main challenges faced in the domain generalization FAS task under a multimodal setting as being reasonable. The experimental results indicate that through effective denoising and alignment strategies, these discrepancies were significantly alleviated, thereby confirming that the proposed denoising and alignment strategies are an effective approach to fundamentally addressing these challenges.

**Missing Modality Scenario During Testing.** In Protocol 2, for each LOO sub-protocol of Protocol 1, we design three test-time missing-modal scenarios to validate the MMDA's performance when modalities are missing. As Table 2 shows, our MMDA framework performs robustly in scenarios with missing modalities, without specific dropout treatments. The average HTER is 10.14%, 11.52% lower than the second-best method, and the average AUC is 95.33%, 10.50% higher. Notably, with the IR modality missing, our method's metrics closely match those without missing modalities, with an AUC of 98.30%. This suggests that the RGB and IR modalities, having learned similar features like lighting and texture, can compensate for each other. However, the absence of the Depth modality

significantly impacts performance, with the **AUC dropping to 93.97%**, highlighting its unique contribution of spatial structural information crucial for accurate identification and classification.

**Limited Source Domain Scenario.** In Protocol 3, we restrict the number of source domains with two sub-protocols, $CW \rightarrow PS$ and $PS \rightarrow CW$. As shown in Table 3, MMDA attains the best results while being trained on only two datasets. Compared with the strongest prior (DADM), MMDA reduces HTER by **5.09 pp** (12.61 % → 7.52 %) and increases AUC by **3.03 pp** (93.81 % → 96.84 %) on **CW→PS**. On **PS→CW**, MMDA reduces HTER by **14.10 pp** (20.40 % → 6.30 %) and increases AUC by **8.84 pp** (89.51 % → 98.35 %). These gains under scarce-source training indicate that MMDA efficiently extracts domain-invariant cues and generalizes across datasets, underscoring its practicality for real-world multimodal FAS.

### 4.2 ABLATION STUDY

**Effectiveness of MD2A.** Table 4 presents a performance comparison of our proposed Domain Differential Attention (MD2A) under two scenarios: Dense

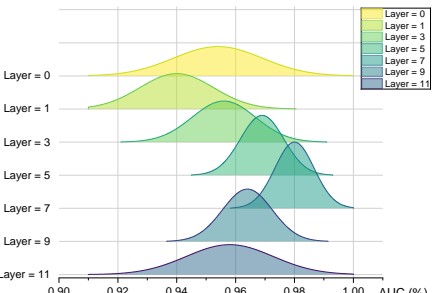

Figure 3: AUC statistics of the U-DSA Module across various caption groups at different depths. The height of each bar represents the number of captions achieving the specified AUC metric. Specifically, this analysis was conducted using a total of ten distinct caption sets to elucidate the impact and distribution of performance metrics at varying depths. This study provides insights into the behavior of the U-DSA at different depth levels and offers valuable perspectives for model optimization.

Adapters and Mixture of Experts (MoE) Adapters, tested under the PS→CW sub-protocol, thereby validating the effectiveness of MD2A and its superiority over traditional Differential Attention (DA). MD2A significantly enhances the model's generalization capability by optimizing domain noise and modality noise. Specifically, without MD2A, the HTER for Dense Adapters was 23.26%, and the AUC was 84.92%. With the introduction of MD2A, the HTER decreased to 13.47%, and the AUC increased to 94.20%. Similarly, for MoE Adapters, the HTER was 22.92% and the AUC was 85.84% without MD2A, but with MD2A, the HTER further decreased to 9.70%, and the AUC increased to 95.23%. These figures indicate that the MoE structure has a stronger capability in learning complex representations and handling data mappings in the representation space. Overall, these results confirm the effectiveness of the denoising strategy and demonstrate the significant role of MD2A in enhancing model performance.

**Effectiveness of RS2.** Table 5 demonstrates the effectiveness of the RS2 alignment method, highlighting the importance of visual representation preservation for generalization. Specifically, the conventional alignment method had an HTER of 9.70% and an AUC of 95.23%. Smooth alignment reduced the HTER to 9.17% and increased the AUC to 96.32%. Notably, the RS2 method further optimized these results, reducing the HTER to 8.88% and achieving an AUC of 97.20%. These results indicate that the RS2 method significantly preserves generalizable representations, which is crucial for excellent performance in multimodal domain generalization for facial anti-spoofing tasks.

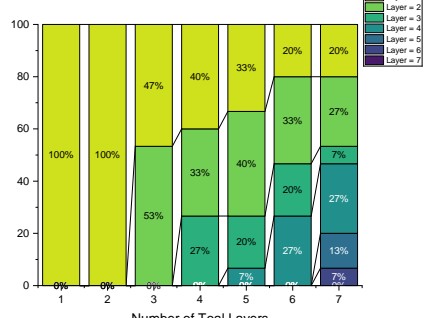

Figure 4: A visualization of the statistics of the layers achieving the best alignment effects in different representation spaces constructed by U-DSA under various total layer numbers (1 to 7 layers).

**Effectiveness of U-DSA.** Fig. 3 shows the ablation results of the U-DSA module in the MMDA framework, highlighting how different layer counts affect performance. We utilized different caption sets to construct various representation spaces. For specific examples, please refer to Appendix A. The U-DSA module primarily boosts the framework's adaptability to various representation spaces. Without it (zero layers), the framework struggles with complex spaces, showing limited generalization. However, adding the U-DSA module, especially at 7 layers, markedly enhances adaptability and generalization across tested spaces, underscoring its importance in managing complex domains. These findings support the value of retaining and adjusting representation distributions for improved module generalization.

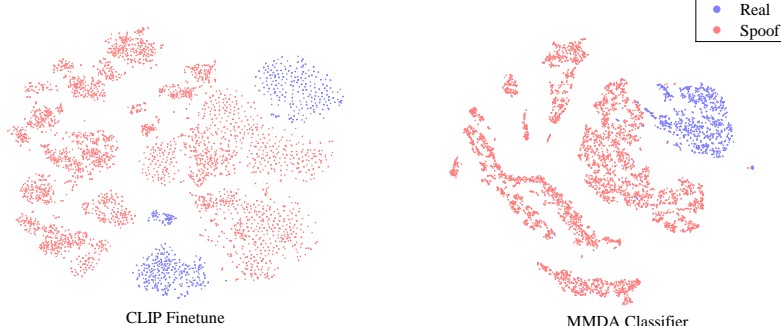

CLIP Finetune          MMDA Classifier

Figure 5: t-SNE visualization of the fine-tuned CLIP (left) and the classifier part of MMDA (right).

Furthermore, as shown in Fig. 4, we conducted an ablation study on the alignment performance of the U-DSA module at different layer depths. The study found that as the total number of layers increases, the optimal performance metrics are mostly concentrated in the shallower layers. This phenomenon indicates that the representation space constructed using pre-training has significant generalization advantages. Notably, in all tested total layer numbers, the deepest layer never achieved the best performance metrics. This suggests that relying solely on representation adjustment and adaptation is insufficient, leading to generalization deficiencies. It also highlights the necessity of remapping operations to ensure that the model can maintain generalization performance by leveraging the generalized representation space.

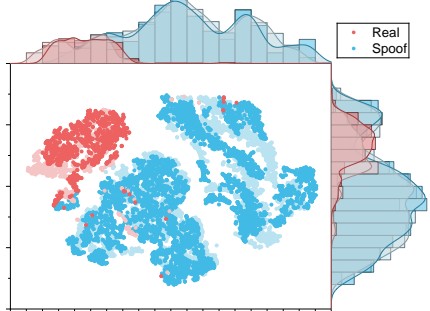

Figure 6: The t-SNE visualization of the U-DSA module is presented, illustrating the data distribution at layer 0 and layer 14. The lighter the color, the closer the data is to layer 0; the darker the color, the closer it is to layer 14.

### 4.3 VISUALIZATION AND ANALYSIS

The t-SNE visualization results in Fig. 5 clearly demonstrate the advantages of the MMDA framework over CLIP in terms of data representation. The representations generated by the MMDA model are more concentrated and consistent, indicating higher distinguishability and coherence among the data. This enhanced representation capability directly reflects MMDA's superior modeling approach, enabling it to more effectively discern the features of genuine and spoof samples.

Figure 6 further reveals the adaptation effect of the U-DSA module in MMDA on the data. The visualization shows the distribution of the data from before entering the U-DSA module (lighter color) to after the remapping is completed and the data output from the U-DSA module (darker color). It can be seen that the representations after adaptation by the U-DSA module gradually become more compact. This gradual transition towards tighter clustering highlights the crucial role of deep alignment in refining feature extraction and representation adaptation. By maintaining and enhancing the generalizability of representations through deeper processing, MMDA demonstrates its unique strengths in feature extraction.

## 5 CONCLUSION

We presented the Multi Modal Denoise and Alignment (MMDA) framework for generalized multimodal face anti spoofing. Across comprehensive experiments and ablations, MMDA improves cross domain performance under complete, missing, and limited source settings. The core components, Modality–Domain Joint Differential Attention (MD2A), the RS2 alignment strategy, and the UDSA module, work together to enhance robustness and generalization.

Limitations remain. A mixture of experts backbone can be sensitive during optimization, and strong appearance changes such as tattoos or heavy makeup may still induce false positives. Future work will stabilize mixture of experts training, further refine UDSA, and extend MMDA toward a unified treatment of physical and digital threats, including caption guided generalization. For a focused discussion of limitations and planned extensions, see Appendix G.

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

## A    CAPTION EXAMPLES

Table 6 presents one of the sets of captions used to construct the representation space. These captions cover descriptions of various attack types and real faces.

## B    COMPARISON OF DIFFERENT CLASSIFICATION METHODS

The MMDA framework's architecture supports a zero-shot classification approach similar to CLIP, and we will explore its comparison with the classification method using a classifier. These two classification methods differ in their implementation during testing. For the classifier-based method, no text input is required during testing; instead, the image is fed into MMDA and processed through the classifier to obtain the final result. In contrast, the zero-shot classification method requires text input. After the image and text inputs are processed by the U-DSA module, the similarity between the image embedding and the text embeddings representing the true and false labels is computed to determine the final classification result.

The detailed comparison is shown in Table 7. In terms of performance, the differences between the two methods are negligible, likely because the performance of both zero-shot classification and the

Table 6: A set of Caption examples.

| Spoof | Real |
|---|---|
| 'This is a photo of a spoof face' | 'This is a photo of a real face' |
| 'This is a photo of an attack face' | 'This is a photo of an authentic face' |
| 'This is a photo of a fake face' | 'This is a photo of a vital face' |
| 'This is a photo of a non-real face' | 'This is a photo of a biometric face' |
| 'This is a photo of a counterfeit face' | 'This is a photo of a genuine face' |
| 'This is a photo of a printed spoof face' | 'This is a photo of an alive face' |
| 'This is a photo of an adversarial patch face' | 'This is a photo of a real-time face' |
| 'This is a photo of a wearable adversarial sticker face' | 'This is a photo of a vivacious face' |
| 'This is a photo of a printed adversarial image face' | 'This is a photo of a veritable face' |
| 'This is a photo of a physical impersonation attack face' | 'This is a photo of a valid face' |
| 'This is a photo of an adversarial perturbation face' | 'This is a photo of an actual face' |
| 'This is a photo of an adversarial makeup face' | 'This is a photo of an existent face' |
| 'This is a photo of a facial attribute modification face' | 'This is a photo of a true face' |
| 'This is a photo of an adversarial face image' | 'This is a photo of a legitimate face' |
| 'This is a photo of a physical-world printed attack face' | 'This is a photo of a natural face' |
| 'This is a photo of a synthetic image face' | 'This is a photo of an unfeigned face' |
| 'This is a photo of an adversarial face mask' | 'This is a photo of an unimpeachable face' |
| 'This is a photo of a semantic adversarial attack face' | 'This is a photo of a credible face' |
| 'This is an example of a spoof face' | 'This is an example of a real face' |
| 'This is an example of an attack face' | 'This is a bonafide face' |
| 'This is not a real face' | 'This is a real face' |
| 'This is how a spoof face looks like' | 'This is how a real face looks like' |
| 'a photo of a spoof face' | 'a photo of a real face' |
| 'a printout shown to be a spoof face' | 'This is not a spoof face' |

Table 7: Comparison of different classification methods during testing.

| Method | Performance | |
|---|---|---|
| | HTER(%)↓ | AUC(%)↑ |
| Zero-shot Classification | 4.12 | 98.23 |
| Classifier-based Classification | 4.00 | 98.94 |

classifier is determined by the alignment achieved by the U-DSA module. Considering computational cost and stability, we recommend the classifier-based classification method. This method does not require text input processing, resulting in lower computational overhead and greater stability.

## C   COMPARISON OF DIFFERENT CLIP BACKBONE METHOD

We compared the performance metrics under different backbones to verify the superiority of CLIP. As shown in Table 8, compared with using ViT as the backbone, the performance of using CLIP as the backbone has been significantly improved, which demonstrates that CLIP has a significant advantage in feature extraction.

Table 8: The results of comparing different CLIP backbones

| Method | CPS→W | | CPW→S | | CSW→P | | PSW→C | | Average | |
|---|---|---|---|---|---|---|---|---|---|---|
| | HTER(%)↓ | AUC(%)↑ | HTER(%)↓ | AUC(%)↑ | HTER(%)↓ | AUC(%)↑ | HTER(%)↓ | AUC(%)↑ | HTER(%)↓ | AUC(%)↑ |
| ViT Backbone | | | | | | | | | | |
| CMFL + ViT George & Marcel (2021) | 22.14 | 84.79 | 28.22 | 79.60 | 24.59 | 82.10 | 39.91 | 62.57 | 28.72 | 77.27 |
| SSDG + ViT Jia et al. (2020) | 25.50 | 82.93 | 28.64 | 78.68 | 29.10 | 76.89 | 36.85 | 67.01 | 30.02 | 76.38 |
| SSAN + ViT Wang et al. (2022) | 22.10 | 87.04 | 22.16 | 86.33 | 28.88 | 75.20 | 45.08 | 56.92 | 29.56 | 76.37 |
| IADG + ViT Zhou et al. (2023) | 17.36 | 90.41 | 18.79 | 89.03 | 25.16 | 82.31 | 35.60 | 68.42 | 24.23 | 82.54 |
| CLIP Backbone | | | | | | | | | | |
| SSDG + CLIP Jia et al. (2020) | 9.09 | 97.14 | 11.17 | 95.25 | 16.19 | 92.64 | 26.25 | 82.72 | 15.68 | 91.94 |
| SSAN + CLIP Wang et al. (2022) | 13.77 | 94.43 | 10.11 | 96.32 | 16.63 | 93.00 | 24.92 | 83.55 | 16.36 | 91.83 |
| IADG + CLIP Zhou et al. (2023) | 14.2 | 93.47 | 9.77 | 94.69 | 22.18 | 85.24 | 23.21 | 83.24 | 17.34 | 89.16 |
| CLIP Radford et al. (2021) | 14.55 | 90.47 | 18.17 | 90.02 | 24.13 | 83.15 | 38.33 | 65.71 | 24.63 | 83.00 |
| **MMDA (Ours)** | **1.22** | **99.99** | **4.21** | **98.62** | **4.34** | **98.58** | **6.25** | **98.18** | **4.00** | **98.94** |

## D   COMPUTATIONAL COST

Table 9 reports complexity and runtime on a single RTX 3090 Ti under the complete–modality setting (RGB+IR+Depth). All times are forward-pass latency *per batch* (batch size = 24) at 224×224

without data–loader overhead. FLOPs are theoretical forward FLOPs; parameter counts include adapters and the classifier. For MMDA, inference uses the classifier head and does not invoke the text encoder, ensuring fairness to the CLIP baselines.

Although increasing U-DSA depth adds `Adapt`/`Remap` blocks (hence parameters), the FLOPs growth is modest and the per-batch latency remains comparable to CLIP-based DG methods. Notably, U-DSA(7) achieves the best AUC with only a small latency increase over U-DSA(0).

Table 9: Computational cost comparison

| Method | Inference time | Params. (M) | FLOPs (G) | Training time | AUC (%) |
|---|---|---|---|---|---|
| MMDA (U-DSA Layers=0) | **12.6 ms/batch** | 169.95 | 362.14 | 12.39 min/epoch | 94.83 |
| MMDA (U-DSA Layers=1) | 13.8 ms/batch | 188.88 | 364.25 | 13.21 min/epoch | 95.54 |
| MMDA (U-DSA Layers=3) | 16.4 ms/batch | 226.74 | 368.48 | 14.83 min/epoch | 95.74 |
| MMDA (U-DSA Layers=5) | 19.1 ms/batch | 264.60 | 372.71 | 15.80 min/epoch | 97.63 |
| MMDA (U-DSA Layers=7) | 21.6 ms/batch | 302.46 | 376.94 | 16.76 min/epoch | **98.18** |
| CLIP+SSAN | 31.8 ms/batch | 165.68 | 111.16 | 13.65 min/epoch | 91.83 |
| CLIP+SSDG | 19.1 ms/batch | **150.07** | **100.02** | **10.55 min/epoch** | 91.94 |

# E  DATASETS AND PROTOCOLS

We adopt four public datasets (CeFA, PADISI, SURF, and WMCA) as our training and evaluation corpora. Collected with real sensors under diverse and realistic conditions, these datasets contain bonafide samples and presentation attacks carried out with real world materials (for example printed photos, replay devices, and masks). RGB, IR, and DEPTH modalities are captured in settings that approximate deployment. These datasets are widely used in multimodal presentation attack detection and are regarded as representative of real world conditions. Table 10 summarizes their main characteristics.

Table 10: Summary of the public datasets used in our experiments.

| Dataset | Year | Training (Real/Fake) | Testing (Real/Fake) | Scenarios | Attack Types |
|---|---|---|---|---|---|
| WMCA | 2019 | 1200 / 3600 | 1200 / 5539 | 6 types | Print, replay, occlusion, masks, etc. |
| SURF | 2019 | 900 / 5397 | 1800 / 10764 | 1 type | Print (paper, cloth) |
| PADISI | 2021 | 698 / 598 | 363 / 304 | 1 type | Print, replay, masks, occlusion |
| CeFA | 2020 | 1301 / 1701 | 3600 / 16617 | 3 types | Print, mask, replay |

To thoroughly evaluate the model, we follow the three protocols proposed in MMDG Lin et al. (2024). They assess cross domain performance under different deployment conditions, including complete modalities, missing modalities at test time, and limited source domains.

**Protocol 1: Complete Modalities.**  Multimodal FAS datasets are collected across diverse scenarios and devices, which leads to clear distribution differences. This protocol evaluates the model through four multimodal Leave One Out (LOO) subprotocols built from CeFA (**C**), PADISI (**P**), SURF (**S**), and WMCA (**W**). For example, $CPS \rightarrow W$ means training on **C**, **P**, and **S** and testing on **W**. These settings measure performance on unseen domains.

**Protocol 2: Missing Modalities at Test Time.**  In practical deployments, sensor failures or network issues may cause some modalities to be unavailable. This protocol extends Protocol 1 by covering cases where one or more modalities are missing during testing, including removal of the depth modality, the infrared modality, or both. It evaluates the model's robustness to incomplete inputs.

**Protocol 3: Limited Source Domains.**  Because collecting multimodal FAS data is costly, this protocol restricts the number and size of training datasets to simulate scarce data. It includes two subprotocols that limit the number of source domains: $CW \rightarrow PS$ and $PS \rightarrow CW$. The goal is to test generalization when training data are limited and to verify reliability in real world applications.

## F ABLATIONS AND MODULE DESIGN

### F.1 MODULE ORGANIZATION AND ROLES

We describe three modules within the MMDA framework and summarize their roles and interactions. **MD2A** jointly models and suppresses noise arising from both modality differences and domain shifts. **RS2** performs soft alignment in the representation space, reducing overconfident matching while making use of latent generalization in CLIP. **U-DSA** preserves high level semantics from CLIP and enables residual refinement through deeper layers. The following ablations quantify their effects and interactions.

**Ablation on module interactions.**    We enable or disable modules on a common backbone attention operator (MHSA) and report performance in Table 11. Results indicate that the modules act in a complementary manner.

Table 11: Module interaction ablation with MHSA as the common backbone operator. A check mark indicates the module is enabled. Lower HTER and higher AUC are better.

| MHSA | MD2A | RS2 | U-DSA | HTER (%) ↓ | AUC (%) ↑ |
|:---:|:---:|:---:|:---:|---:|---:|
| ✓ | | | | 14.88 | 92.47 |
| ✓ | ✓ | | | 13.45 | 94.38 |
| ✓ | | | ✓ | 22.08 | 87.43 |
| ✓ | ✓ | | ✓ | 15.65 | 89.77 |
| ✓ | | ✓ | | 9.70 | 95.23 |
| ✓ | ✓ | ✓ | ✓ | 8.88 | 97.20 |

From Table 11, MD2A alone improves over the MHSA baseline, indicating the value of denoising. U-DSA is less effective without prior denoising but improves when combined with MD2A, which suggests synergy. RS2 further strengthens alignment and discrimination. The best results are obtained when all modules are used together.

### F.2 ABLATION ON SKIP CONNECTIONS

We evaluate skip connections in the U-DSA module under the $PSW \rightarrow C$ subprotocol; the results are summarized in Table 12. Adding skip connections yields an AUC increase of $0.51$ and a lower HTER.

Table 12: Effect of skip connections in U-DSA under $PSW \rightarrow C$.

| Method | HTER (%) ↓ | AUC (%) ↑ |
|:---|---:|---:|
| w/o skip connection | 7.02 | 97.67 |
| w/ skip connection | 6.25 | 98.18 |

### F.3 TRANSFERABILITY OF MD2A

MD2A is designed as a modular component that can be inserted into other multimodal architectures with minimal changes. To demonstrate this, we integrate MD2A into a ViT + Adapter architecture and compare against common adapter variants.

As shown in Table 13, incorporating MD2A improves performance and strengthens generalization relative to the alternative adapters. These results suggest that applying MD2A to other architectures, such as vision language models and cross task adapters, is a promising direction for future study.

Table 13: MD2A inserted into ViT + Adapter.

| Method | HTER (%) ↓ | AUC (%) ↑ |
|---|---|---|
| ViT + MHSA Adapter | 15.92 | 91.02 |
| ViT + DA Adapter | 11.42 | 93.76 |
| ViT + MD2A Adapter | 11.22 | 94.46 |

# G LIMITATIONS AND FUTURE WORK

## G.1 LIMITATIONS

MMDA is designed and evaluated for physical presentation attacks in deployment like settings. The main failure mode arises when adversarial appearance changes such as tattoos or heavy makeup create cross modal patterns that resemble bona fide cues, which can lead to false positives; examples are shown in Figure 7.

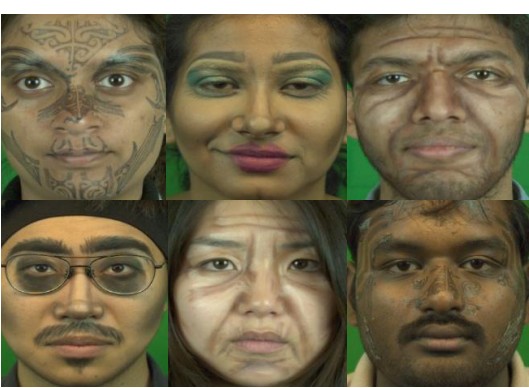

Figure 7: Representative failure cases of MMDA. Tattoo and makeup patterns can exhibit high cross modal consistency and mimic bonafide cues, resulting in false positives. **Note:** All face images are from the public PADISI dataset Rostami et al. (2021).

Table 14: Preliminary cross domain AUC on digital attacks (train on FaceForensics++, test on DFDC).

| Method | AUC (%) ↑ |
|---|---|
| ED (AAAI24) | 72.1 |
| LSDA (CVPR24) | 73.6 |
| FreqDebias (CVPR25) | 74.1 |
| ForensicsAdapter (CVPR25) | 84.3 |
| VB (CVPR25) | 84.3 |
| MMDA | 76.4 |

## G.2 FUTURE WORK

Our study focuses on physical attacks, which align with multimodal FAS deployment scenarios. Looking forward, we will extend MMDA toward a unified treatment of physical and digital threats. Physical attacks present spoof materials to a camera and are detected via live cues and physical inconsistencies, while digital attacks such as deepfakes are generated in virtual environments and are detected via synthesis artifacts or inconsistencies. These families are complementary, not interchangeable.

As a brief feasibility check for digital scenarios, we train on FaceForensics++ and test on DFDC, treating each deepfake type (DeepFake, Face2Face, FaceShifter, NeuralTextures) as a distinct domain. Results are summarized in Table 14. Future extensions will incorporate temporal and audio cues for video based cases, refine cross modal consistency checks to reduce false positives on makeup and tattoo cases, and broaden cross domain evaluation across both physical and digital benchmarks.

