# OpenReview forum: "Denoising and Alignment: Rethinking Domain Generalization for Multimodal Face Anti-Spoofing"
_ICLR.cc/2026/Conference — ICLR 2026 Conference Withdrawn Submission_

### Official Review · Reviewer_MDdK · 2025-10-20

**Soundness:** 2
**Presentation:** 2
**Contribution:** 2
**Rating:** 4
**Confidence:** 5

**Summary:**

This paper introduces the Multimodal Denoising and Alignment (MMDA) framework to enhance domain generalization in multimodal face anti-spoofing. The approach tackles issues of modality bias and domain shift through a combination of denoising and alignment techniques. Specifically, the Modality-Domain Joint Differential Attention (MD2A) module suppresses noise from both modalities and domains during fusion. The Representation Space Soft Alignment (RS2) maps fused representations to text-defined subspaces for better separability, while the U-shaped Dual Space Adaptation (U-DSA) module preserves pretrained semantics through a deep adaptation structure. Extensive evaluations demonstrate that MMDA achieves state-of-the-art results across multiple benchmarks, reducing HTER by 9.63% and increasing AUC by 5.98% under complete modality settings, with robust performance in missing modality and limited source domain scenarios

**Strengths:**

* The paper presents a comprehensive experimental evaluation that effectively validates the effectiveness of MMDA under both protocols. The ablation studies for each component are also meticulous. Furthermore, the manuscript is well-structured and formatted.

**Weaknesses:**

* The motivation of this paper sounds limited. In line 72, the authors combine Figure 1 to explain the two difficulties in the multimodal DG problem of FAS: noise diversity and difficulty in alignment. However, these two difficulties can actually be unified into modality gap and domain gap, which has been mentioned many times in previous papers. Moreover, Figure 1 does not intuitively reflect the impact of modality and domain gap on existing methods.

* The motivations and methodologies underlying Multimodal Domain Alignment (MMDA) are both subjects of ongoing debate. First, why is the noise present across different modalities and domains regarded as the central issue, and what precisely is meant by the term "noise" in this context? Second, within the MMDA framework, samples x_i and x_j originating from the same domain are concatenated at the batch level. However, it remains unclear why the multiplication of the query (Q) and key (K) matrices in Lines 13 and 14 of Algorithm 1 leads to the emergence of noise. This is the point that merits further clarification.

* Figure 2 lacks comprehensiveness in its visualization of all modules. As a result, it is difficult to intuitively discern the architectural compositions and input/output flows of the U-DSA and RS2 modules from the figure alone.

* The rationale behind the concurrent computation of both the classification loss (L_cls) and the alignment loss (L_align) in RS2 is not immediately clear.​​ If the classification loss alone is sufficient for prediction, the specific necessity of the alignment objective remains ambiguous. It would be helpful to understand the distinct role that L_align plays, which is not fulfilled by L_cls.

* In Section 3.3, the description of the U-DSA's design rationale and methodology lacks clarity. The meaning of certain terms remains vague, such as "fine-tuning module," "downstream task," and "pre-training data."

* The presentation of the motivational rationale and technical approach in this paper is not straightforward. This poses a considerable challenge to the reader, who must often guess the intended meaning. Such ambiguity can hinder the reader's understanding, and a thorough restructuring of the exposition is strongly recommended.

**Questions:**

Please refer to the weakness part for detalis.

---

### Official Review · Reviewer_rmYM · 2025-10-27

**Soundness:** 2
**Presentation:** 3
**Contribution:** 2
**Rating:** 4
**Confidence:** 5

**Summary:**

The paper proposes MMDA (Multimodal Denoising and Alignment), a CLIP-based framework for multimodal face anti-spoofing (FAS). It introduces three key components: (1) Modality–Domain Joint Differential Attention (MD2A) for denoising, (2) Representation Space Soft Alignment (RS2) for flexible cross-modal alignment, and (3) U-shaped Dual Space Adaptation (U-DSA) for feature feedback across layers. The method achieves strong empirical performance on multiple datasets and protocols, reporting SOTA results in AUC and HTER under multimodal domain generalization settings.

**Strengths:**

The framework is technically well-structured and empirically validated across multiple benchmarks (CeFA, PADISI, SURF, WMCA).

The ablation studies are detailed, covering all three proposed modules.

Visualization results (t-SNE) and efficiency analysis (Table 9) help make the method transparent and reproducible.

The integration of CLIP-based alignment into multimodal FAS is timely and relevant.

**Weaknesses:**

1. Motivation is not empirically substantiated (two parts).
(a) Figure 1 claims that modality bias makes the IR–Depth gap significantly larger than RGB–RGB, yet the paper provides no dedicated analysis (e.g., inter-modality feature distances, per-modality performance gaps, or distribution visualizations) to support this hypothesis.
(b) The method is argued to “avoid overly smooth decision boundaries”, but the only evidence is a t-SNE plot; there is no direct boundary/margin analysis or causal link showing that sharper boundaries lead to better domain generalizability.

2. Frozen CLIP visual encoder with unseen modalities (IR/Depth) is under-justified and under-specified.
Figure 2 annotates parts of the backbone as “Frozen,” while only the adapter-like modules are marked “Trainable,” implying the CLIP visual encoder is not updated; however, the Implementation Details do not clearly spell out the freezing policy. More importantly, the paper uses infrared and depth inputs, which CLIP was almost certainly not exposed to during pretraining. The authors do not explain how IR/Depth are mapped to CLIP’s RGB input space nor provide ablations comparing frozen vs. unfrozen backbones or LoRA finetune. This choice may hamper low-level feature adaptation to IR/Depth and threatens reproducibility.

**Questions:**

See Weaknesses part.

---

### Official Review · Reviewer_C2br · 2025-10-28

**Soundness:** 3
**Presentation:** 3
**Contribution:** 1
**Rating:** 4
**Confidence:** 4

**Summary:**

This paper addresses the challenge of domain generalization in multimodal face anti-spoofing (FAS), where models often fail under unseen environments or modality biases.

The authors propose MMDA (Multimodal Denoising and Alignment), a framework composed of three modules:

MD2A (Modality–Domain Joint Differential Attention) — removes domain and modality noise via a differential attention mechanism using same-domain sample pairs.

RS2 (Representation Space Soft Alignment) — aligns visual embeddings to text-defined class subspaces (rather than a single prompt), leveraging CLIP’s multimodal representation.

U-DSA (U-shaped Dual Space Adaptation) — performs layer-wise feedback and multi-level alignment to preserve generalizable semantics while adapting to downstream FAS tasks.

The paper claims substantial performance gains over previous methods on four multimodal FAS benchmarks.

**Strengths:**

- Proposed a conceptually unified pipeline combining denoising, alignment, and adaptation. To address an important and practical problem: robust multimodal FAS under domain shift.

- Module MD2A introduced a differential attention mechanism to deal with modality and domain biases.

- Module RS2’s flexible text-subspace alignment is novel and intuitively appealing for CLIP-based multimodal tasks.

- Comprehensive experiments on multiple datasets and settings.

**Weaknesses:**

- Questionable theoretical justification of MD2A:

The claim that same-domain sample pairs isolate “noise” is not empirically proven; cross-domain comparisons or explicit visualization of noise components are missing. The denoising term appears more heuristic than theoretically motivated.

- Insufficient clarity and ablation for U-DSA:

The U-shaped Dual Space Adaptation is conceptually interesting but underexplained. The mechanism of “Remap” and its interaction with layer-wise RS2 losses are not fully detailed.

- Lack of interpretability analysis:

Although the paper positions MMDA as improving robustness through “denoising and alignment,” there is no qualitative analysis (e.g., feature visualization, attention heatmaps) to confirm that noise suppression or semantic alignment occurs as intended. This weakens the causal link between the method’s motivation and its empirical outcomes.

**Questions:**

Can you empirically verify that same-domain differential attention indeed captures noise?
For example, visualize the attention difference maps or compare with cross-domain differential attention.

How sensitive are the results to the λ parameter controlling noise suppression in Eq. (2)?

How is “Remap” implemented in U-DSA? Are weights shared across layers?
Is the RS2 loss applied at every layer with equal weight, or only at selected depths?

Have you evaluated robustness to unseen attack types beyond dataset-level domain shifts?

---

### Official Review · Reviewer_F79X · 2025-11-02

**Soundness:** 3
**Presentation:** 3
**Contribution:** 2
**Rating:** 4
**Confidence:** 5

**Summary:**

The paper proposes a training method called MMDA designed to help spoof face detectors generalize against unseen attacks in multimodal settings. Specifically, to mitigate domain-wise noise, they implement the Differential Attention MD2A, which produces cleaner representations. The soft alignment objective at the output end aligns the visual modality representation with the closest corresponding text prompt. Various experiments are conducted, demonstrating the effectiveness of MMDA across different setting protocols.

**Strengths:**

- The paper applied various techniques for aligning pre-trained space on source dataset and pushing modalities representation more compact.

- All modules in the paper are well motivated and explained, with acceptable illustration.

- The proposed MMDA method have shows its superior performance across evaluation protocol: unseen domain testing, limited training source domain, and missing modalities scenarios.

**Weaknesses:**

- The contribution of the U-shaped Dual Space Adaptation Module (U-DSA) is not well-studied. Specifically, the ablation study in Table 11 indicates that performance degrades whenever U-DSA is included (for example, comparing line 1 vs. line 3, and line 2 vs. line 4). The ablation study in Figure 3 solely shows the effect of the number of layers on output performance, not a comparison against a simpler alternative, such as using only MLP layers (Adapt). Meanwhile, the dilemma of deep versus shallow structure selection mentioned in lines 304–306 could be solved with a simpler residual addition (e.g., $v_i = v_{i-1} + \text{Adapt}(v_{i-1})$). Additionally, there is no distinct effect of U-DSA visible in Figure 6, suggesting a need for a statistical significance test to numerically expose its benefit. Lastly, the authors study the effect of a 'skip connection' in Table 12 (Section F2), but 'skip connection' is never mentioned in Section 3.3.

- The Modality-Domain Joint Differential Attention (MD²A) module is not properly attributed in Section 3.2. The author's writing makes it seem like it is their own proposed novel method. Specifically, Differential Attention was introduced by Tianzhu Ye in Differential Transformer in ICLR 2025.

- There is a wrong statement in line 172: SA-FAS by Sun et al. (2023) does not apply a parameter-efficient method; they simply train the whole ResNet18 on the source domain datasets.

- The citation format in the paper is incorrect according to English grammar. For instance, in Line 37, 'Yu et al. (2022); Xu et al. (2023)' should be placed inside parentheses. - Several typos need to be fixed, such as in Line 215: 'Alg. 1,' should be changed to 'Alg. 1,'.

**Questions:**

- What is the significant and distinct effect of the U-DSA Module? Can it be simply replaced by a residual addition? If so, could the authors include an ablation study comparing U-DSA against this alternative, across a varying number of layers?

- How is the proposed Modality-Domain Joint Differential Attention (MD2A) module distinct from the Differential Transformer proposed by Ye et al. (ICLR 2025)?

- Why $L_{align}$ targets on soft alignment but not using the farthest-anchor cosine distance, as it helps modality representation be more compact.

- In the experiments, did the authors report snapshot performance or final epoch performance? If a snapshot was reported (e.g., the best performance achieved during training), could the authors provide an analysis of the algorithm's convergence behavior in the last few epochs?

- How well does the MMDA method generalize when tested on traditional unimodal datasets (such as MSU-MFSD, CASIA-FASD, Idiap ReplayAttack, and OULU-NPU)?

---

### Note · Authors · 2025-11-12

**Comment:**

We sincerely thank all reviewers for their careful evaluations and constructive feedback on our submission. After carefully reading and discussing the reviews, we realize that the current version of the paper requires revision. Therefore, we have decided to withdraw this submission and thoroughly revise this paper by fully incorporating the reviewers’ comments. We are very grateful to the reviewers and chairs for their time and effort.

**Withdrawal Confirmation:**

I have read and agree with the venue's withdrawal policy on behalf of myself and my co-authors.